# Investigation on UV Degradation and Mechanism of 6:2 Fluorotelomer Sulfonamide Alkyl Betaine, Based on Model Compound Perfluorooctanoic Acid

Naveed Ahmed [1,2], Marion Martienssen [2], Isaac Mbir Bryant [3], Davide Vione [1], Maria Concetta Bruzzoniti [1] and Ramona Riedel [2,*]

1   Department of Chemistry, University of Turin, Via Pietro Giuria 5, 10125 Turin, Italy; naveed.ahmed@unito.it (N.A.); davide.vione@unito.it (D.V.); mariaconcetta.bruzzoniti@unito.it (M.C.B.)
2   Chair of Biotechnology of Water Treatment, Institute of Environmental Technology, Brandenburg University of Technology Cottbus-Senftenberg, 03046 Cottbus, Germany; martiens@b-tu.de
3   Department of Environmental Science, University of Cape Coast, Cape Coast 4P4872, Ghana; ibryant@ucc.edu.gh
*   Correspondence: ramona.riedel@b-tu.de

**Abstract:** The UV treatment of 6:2 FTAB involves the mitigation of this persistent chemical by the impact of ultraviolet radiation, which is known for its resistance to environmental breakdown. UV treatment of PFOA and/or 6:2 FTAB, and the role of responsible species and their mechanism have been presented. Our investigation focused on the degradation of perfluorooctanoic acid (PFOA) and 6:2 fluorotelomer sulfonamide alkyl betaine (6:2 FTAB, Capstone B), using UV photolysis under various pH conditions. Initially, we used PFOA as a reference, finding a 90% decomposition after 360 min at the original (unadjusted) pH 5.6, with a decomposition rate constant of $(1.08 \pm 0.30) \times 10^{-4}$ sec$^{-1}$ and a half-life of $107 \pm 2$ min. At pH 4 and 7, degradation averaged 85% and 80%, respectively, while at pH 10, it reduced to 57%. For 6:2 FTAB at its natural pH 6.5, almost complete decomposition occurred. The primary UV transformation product was identified as 6:2 fluorotelomer sulfonic acid (6:2 FTSA), occasionally accompanied by shorter-chain perfluoroalkyl acids (PFAAs) including PFHpA, PFHxA, and PFPeA. Interestingly, the overall decomposition percentages were unaffected by pH for 6:2 FTAB, though pH influenced rate constants and half-lives. In PFOA degradation, direct photolysis and reaction with hydrated electrons were presumed mechanisms, excluding the involvement of hydroxyl radicals. The role of superoxide radicals remains uncertain. For 6:2 FTAB, both direct and indirect photolysis were observed, with potential involvement of hydroxyl, superoxide radicals, and/or other reactive oxygen species (ROS). Clarification is needed regarding the role of $e_{aq}^-$ in the degradation of 6:2 FTAB.

**Keywords:** PFOA; 6:2 FTAB; decomposition; photolysis; scavenger

## 1. Introduction

Per- and poly-fluoroalkyl substances (PFASs) represent a class of anthropogenic compounds that raise serious health and ecological concerns due to their toxicity and environmental persistence [1–3]. PFASs are extensively used in a wide range of industrial and commercial applications including paints, cookware, water-repellent textiles, alkaline cleaners, packaged foods, carpets, upholstery, shampoos, and firefighting foams [4,5]. From PFAS-containing products, these substances are discharged into industrial and municipal wastewater, leading to their ubiquitous occurrence in different environmental compartments including wildlife tissues, human blood, and breast milk [6].

Recently, several thousand individual PFASs have been employed. Many of these substances contain perfluorooctanoic acid (PFOA) and perfluorooctane sulfonate (PFOS) as backbone structures [7]. They have non-enforceable lifetime health advisory levels set at

70 parts per trillion (ppt) by the U.S. Environmental Protection Agency (US EPA) in 2016 [6]. PFOA has been associated with adverse health effects, including potential links to cancer, reproductive issues, and impacts on the liver, while also posing environmental concerns due to its bioaccumulation and long-lasting presence in ecosystems [8].

The use of PFAS-containing aqueous film-forming foams (AFFFs) for extinguishing fires caused by hydrocarbon fuels in places like airports, offshore platforms, and industrial buildings has become a serious environmental issue. For instance, the PFOS alternative 6:2 fluorotelomer sulfonamide alkyl betaine (6:2 FTAB) is a common ingredient of AFFF [9]. Only little is known about its toxicity, even if Shi et al. [9] suggested that 6:2 FTAB seems to be less toxic than PFOS. Unfortunately, 6:2 FTAB is often found as a widespread pollutant in surface water [10], groundwater [11], soil [12], and other environments. According to Houtz et al. [13], precursor chemicals of perfluoroalkyl acids (PFAAs) found in AFFFs are gradually degraded into PFAAs and linger in the environment for decades [14]. Unlike PFAAs, fluorotelomer-based chemicals have exchangeable hydrogen atoms and comparatively electron-rich carbon atoms, making them more susceptible to environmental oxidation [15,16]. This feature reduces their persistence and, thereby, limits their bioaccumulation potential. However, fluorotelomer-based chemicals may undergo environmental decomposition and their by-products might persist for a long time. Harding-Marjanovic et al. [17] and Weiner et al. [18] found that the 6:2 fluorotelomer mercaptoalkylamido sulfonate (FTSAS) and its 4:2 and 8:2 analogues biodegrade to produce perfluoroalkyl acids (PFAAs). More relevant to environmental conditions, Gauthier and Mabury [19] reported the oxidation of 8:2 fluorotelomer alcohol (8:2 FtOH) initiated by hydroxyl radical ($^\bullet$OH) in irradiated hydrogen peroxide ($H_2O_2$) solutions, in both synthetic and natural lake water, leading to PFAA products. In this work, we therefore selected PFOA as a model compound because shorter-chain PFAAs are released from PFOA and fluorotelomer-based PFASs.

Much research has been carried out investigating advanced oxidation processes (AOP) to break PFASs down [20]. Overall, two decomposition pathways of ultra-violet (UV) irradiated PFOA are reported in the recent literature: (i) direct decomposition and (ii) indirect decomposition. Direct photolysis of PFOA typically occurs in the UV range below 220 nm and it has a higher decomposition efficiency if vacuum UV (VUV, ≤185 nm) is applied for treatment. For example, Chen et al. [21] found very slow PFOA decomposition through direct photolysis under irradiation at 254 nm, and considerably faster decomposition under 185 nm irradiation. The PFOA molecule strongly absorbs UV light from 190 to 220 nm, leading to decarboxylation [21]. However, the application of VUV is not representative for environmental conditions and, therefore, prediction of environmental decomposition mechanisms might not be completely transferable.

The indirect decomposition pathway is basically radical-driven. Several studies have reported that $^\bullet$OH and superoxide radicals ($O_2{}^{\bullet-}$) are involved in the decomposition of PFOA [22–24], although a minority of scientists have opposed the idea of their involvement [24–29]. However, it is generally proposed that the involved radical(s) might first break down the C-C bonds (the same might be performed by UV radiation alone, however) and form PFOA radicals ($^\bullet C_7F_{15}$) through photo-Kolbe decarboxylation. The PFOA radical ($^\bullet C_7F_{15}$) in turn reacts with $^\bullet$OH to form a perfluorinated alcohol intermediate, $C_7F_{15}OH$ [30]. The latter is not stable and ultimately releases $CF_2$ units, further forming $C_6F_{13}COOH$ via subsequent hydrolysis [23–25,31–36]. Other scientists claimed that the UV decomposition of PFOA is driven by hydrated electrons ($e_{aq}^-$) [37–39]. The UV photolysis produces the following species, as given in Equations (1) and (2) [40].

$$H_2O + h\nu \ (190 \ nm) \rightarrow H^\bullet + {}^\bullet OH \tag{1}$$

$$H_2O + h\nu \ (190 \ nm) \rightarrow H^+ + {}^\bullet OH + \tag{2}$$

Overall, despite different views on the subject, there is general agreement that the indirect decomposition mechanism of PFOA is very complex and depends at least on the substrate

concentration, pH, irradiation source and photo flux, presence of dissolved or molecular oxygen, and addition of further reaction partners (i.e., potential catalysts) [21,37–39,41]. In particular, the effect of pH on the decomposition of PFASs should not be neglected. Giri et al. [41] have investigated five important parameters influencing the photochemical decomposition of PFOA. They stated that the decomposition was enhanced at acidic pH, as far as both treatment time and decomposition efficiency were concerned. Furthermore, they found that dissolved oxygen had a negative impact on PFOA degradation at pH 5.5. Wang and Zhang [37] confirmed that dissolved oxygen inhibited the decomposition of PFOA. Similar results for PFOS decomposition were reported by Jin et al. [42], showing rapid PFOS decomposition in anaerobic alkaline conditions under monochromatic 185 nm irradiation. The latter authors also found that photochemical PFOS decomposition was faster under anaerobic alkaline than under aerobic neutral conditions. The decomposition rates of branched PFOS isomers were much faster than those of linear PFOSs ($c_o$ = 10.0 mg L$^{-1}$; k = 0.0806 min$^{-1}$), indicating that $e_{aq}^-$ may be the active species in the process [43].

Early studies of PFAS photolysis have not received much attention, because they demonstrated that the majority of PFASs undergo very slow or even negligible degradation under UV [21,30,43]. Researchers then solved this problem by introducing photo-sensitisers during photodegradation, to achieve higher decomposition efficiency and a broader absorption band [29,44–46]. By introducing trace amounts of ferric ions (0–5.0 mM), Hori et al. [47] improved the photochemical decomposition of 67.3 mM PFAA by 3.6 times under 254 nm radiation. Furthermore, there have been relatively few studies into the removal of other PFASs besides PFOA and PFOS and, especially, regarding long-chain telomers such as 6:2 FTAB.

This article clarifies the ambiguity of PFOA decomposition through superoxide radicals, and the 6:2 FTAB degradation mechanism through the involvement of reactive oxygenated species (ROS). For the above reasons, the aim of our study was to investigate the photolysis of aqueous 6:2 FTAB at different pH values (4, 7, and 10). We already studied the UV decomposition of PFOA as a model compound, to prove whether our UV system produces similar results as compared with the recent literature. Subsequently, we applied the same treatment conditions to 6:2 FTAB. We determined the decomposition kinetics and the release of major transformation products (TPs) by LC-MS analysis. We also investigated the potential involvement of radicals mediating the decomposition through scavenger experiments. Based on our results, we finally propose a decomposition mechanism for the UV photolysis of 6:2 FTAB.

## 2. Materials and Methods

### 2.1. Chemicals

PFOA was purchased from Acros Organics (USA). 6:2 Fluorotelomer sulfonamide alkyl betain (6:2 FTAB) was purchased as Capstone 1157 from Chemours (Meyrin, Switzerland). Methanol ($CH_3OH$) and ethanol ($C_2H_5OH$) were purchased from Merck (Darmstadt, Germany), and 2-propanol ($C_3H_7OH$) from VWR (Darmstadt, Germany). Butanol ($C_4H_9OH$), iron trichloride ($FeCl_3 \cdot 6H_2O$), and formic acid (HCOOH) were purchased from Merck (Darmstadt, Germany). L-Threoascorbic acid was purchased from VWR (Leuven, Belgium). $H_2SO_4$ was purchased from Roth (Karlsruhe, Germany) and LC-MS grade acetonitrile from VWR (Dresden, Germany). All chemicals were of analytical grade or better, with purity > 99%. Ultrapure water (LC-MS grade) was generated in-house (Adrona Sia Crytal EX, Riga, Lithuania).

### 2.2. Experimental Set-Up and Photolysis of PFOA and 6:2 FTAB

In our UV experiments, a 150 W medium pressure mercury lamp (TQ 150, Heraeus Noblelight, Hanau, Germany) was used, which has a wavelength range from 190 to 600 nm and an emission maximum at 366 nm. The system configuration has been described earlier as open system configuration 1 (open quartz-glass vessel) with outer cooling jacket, Kuhn et al. [48], Figure S1. This device produced an incident photon flux of (3.81 ± 0.19) ×

$10^{-5}$ E sec$^{-1}$, which corresponds to a light intensity of $1.24 \pm 0.1$ mW cm$^{-2}$ (according to actinometry results, see Figure S2). A magnetic stirrer was used for continuous mixing of the solution. The pH of the treatment solutions was adjusted by adding either ammonia or formic acid prior to the UV treatment. The temperature in the reactor was kept between 20 and 25 °C throughout the experiment. All UV treatments were carried out using either PFOA or 6:2 FTAB (1 mg L$^{-1}$) in a total reactor volume of 320 mL. The UV lamp was turned on 10 to 15 min before each experiment. Samples (5 mL) were collected at t = 0 and subsequently at 15 min intervals for 180 min, then at 30 min intervals for further 180 min (i.e., total treatment time 360 min).

### 2.3. Radical Scavenging Experiments

Scavenger experiments were carried out with either PFOA (pH 5.6) or 6:2 FTAB (pH 6.5). A concentration of 0.3 M alcohols or 0.3 mM ascorbic acid was used to scavenge reactive species and, whenever required, the scavenger was added before the UV treatment. Samples were collected at the starting point and, afterwards, at 15 min intervals for 180 min followed by 60 min intervals up to 180 min for a total of 360 min (180 min + 180 min). All tests were performed in triplicates. Liquid samples were analysed by LC-MS for either PFOA or 6:2 FTAB and their TPs. The release of fluoride and sulfate ions was analysed by ion chromatography (IC).

### 2.4. Quantification of PFOA, 6:2 FTAB, and Their Metabolites by LC-MS

PFOA and potential TPs were identified and quantified samples were analysed by liquid chromatography-electron spray ionisation-mass spectrometry (LC-ESI-MS) using an LC-MS-IT-TOF LC-MS system from Shimadzu (Tokyo, Japan), equipped with (Shimadzu) LC Spectral System P4000, LCQ MS Detector, and Autosampler AS 3000. A 10 µL aliquot of each sample was injected into the LC system. The sample was separated using an Ultra AQ C18 column (150 × 2.1 mm, 3 µm/200 Å Restek, Bellefonte, PA, USA). The column temperature was 35 °C and the flow rate was constant at 0.2 mL min$^{-1}$. Linear gradient elution was performed with solvent A (0.1% formic acid in ultra-pure water) and solvent B (0.1% formic acid in acetonitrile). The analysis was run for 35 min with the following gradient: 20% B for 3 min, then up to 60% B in 1 min and held for 15 min, then up to 80% B in 1 min and held for a further 5 min, and finally back to 20% B in 3 min and held for another 7 min. The MS detector settings were as follows: negative ionisation at 1.7 kV, capillary spray temperature of 220 °C, and 10 msec ion accumulation time. In scan mode, the mass-to-charge ratio (m/z) was measured in the 150–750 range. In SIM mode, the following *m/z* ratios were measured: 569.0785, 426.9742, 412.9682, 362.9699, 312.9731, 262.9762, and 212.9887, corresponding to 6:2 FTAB, 6:2 FTSA, PFOA, PFHpA, PFHxA, PFPeA, and PFBA, respectively.

For quantification, sample volumes of 1 mL were mixed with 10 µL internal standard (13C8 PFOA/13C8 PFOS, Cambride Isotopes, Andover, MA, USA). Calibration was done with the multi-component standard ITA-70 (5 to 200 µg L$^{-1}$, Agilent Technologies, Santa Clara, CA, USA).

### 2.5. Fluoride and Sulfate Measurement by Ion Chromatography

UV-treated samples were analysed for fluoride release using a DX-120 ion chromatograph (IC) (Dionex, Sunnyvale, CA, USA) equipped with an IonPac AS22 column (250 × 4 mm, Dionex, Sunnyvale, CA, USA) and a Dionex ASRS 4 mm membrane suppressor for suppressed ion conductivity detection. A total volume of 20 µL was injected into the IC system by using autosampler Gynkoteck GINA 50 (Germering, Germany) equipped with a 20 µL loop, using 4.5 mM NaCO$_3$ and 1.4 mM NaHCO$_3$ as the eluent. Fluoride and sulfate were eluted at retention times of 2.8 min and 11.9 min, respectively.

*2.6. Dissolved Oxygen and Hydrogen Peroxide Measurements*

Preliminary experiments were performed to monitor the consumption of dissolved oxygen (DO) and release of $H_2O_2$ during UV treatment of PFOA. DO was measured by an optical dissolved oxygen sensor using Fibox 3 minisensor oxygen meter from PreSens (Regensburg, Germany). Potential release of $H_2O_2$ was measured by the DPD (N,N-diethyl-p-phenylenediamine) method.

## 3. Results and Discussion

*3.1. Decomposition of PFOA at Different pH Values*

We investigated the influence of three different pH values on the decomposition kinetics and mechanism of UV-treated PFOA, and on TPs formation. Among many reported factors such as temperature, dosage of oxidant, or dissolved oxygen, pH plays a crucial role because it affects both the generation rate of radicals and the speciation of many contaminants. Both radicals formation and contaminant speciation highly affect the decomposition performance [49,50].

In our experiments, PFOA (1 mgL$^{-1}$) in aqueous solution had a pH of 5.6 and degraded rapidly during UV irradiation (Figure 1). After 360 min treatment, 90.3 ± 2.5% of PFOA was degraded. Assuming a pseudo-first-order kinetics (Figure 1b) produced by the involvement of one or more reactive species in PFOA decomposition, we determined a half-life of $t_{1/2}^{\text{PFOA}}$ = 106.8 ± 2.0 min (Table 1). Interestingly, PFOA decomposition percentages at pH 4.0 and pH 7.0 were slightly lower as compared with pH 5.6, 85.2 ± 1.6%, and 80.5 ± 4.0%, respectively (Figure 1a,c). However, the calculated half-lives were significantly different (Table 1), with $t_{1/2}^{\text{PFOA}}$ = 130.4 ± 2.4 min at pH 4.0 and 152.4 ± 2.9 min at pH 7.0. In addition, we determined a significant decrease in the PFOA decomposition at pH 10.0 that averaged 57.9 ± 1.8%, with $t_{1/2}^{\text{PFOA}}$ = 288.4 ± 2.6 min (Figure 1d). Thus, the highest decomposition and shortest half-life was determined at the natural pH of aqueous PFOA. Compared to pH 5.6, the decomposition rate constants at pH 4.0, 7.0, and 10.0 were 0.82, 0.70, and 0.37, respectively.

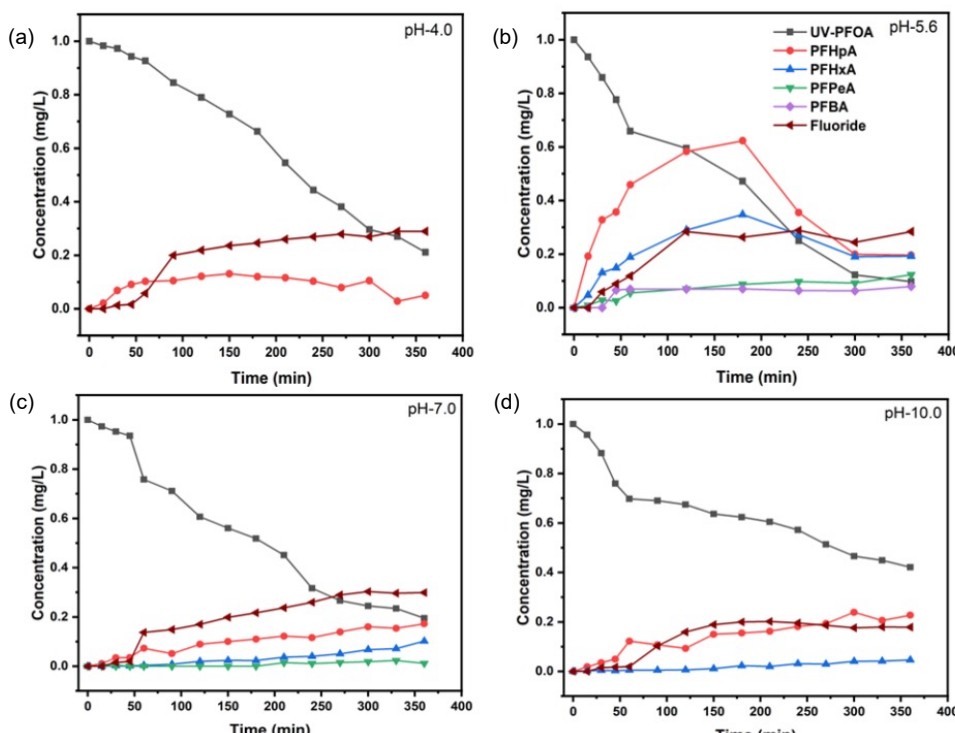

**Figure 1.** Removal of PFOA, production of transformation products, and fluoride release upon UV irradiation at pH 4.0 (**a**), pH 5.6 (**b**), pH 7.0 (**c**), and pH 10.0 (**d**).

**Table 1.** Rate constant, half-life, %age decomposition, and defluorination at different pH values for 1 mg L$^{-1}$ PFOA.

| pH Value | k (s$^{-1}$) | t$_{1/2}$ (min) | Decomposition (%) | Fluoride Release (%) |
|---|---|---|---|---|
| 4.0 | $(8.85 \pm 1.13) \times 10^{-5}$ | $130 \pm 2$ | $85.2 \pm 1.6$ | $29.0 \pm 3.5$ |
| 5.6 | $(1.08 \pm 0.30) \times 10^{-4}$ | $107 \pm 2$ | $90.3 \pm 2.5$ | $28.8 \pm 4.8$ |
| 7.0 | $(7.57 \pm 1.70) \times 10^{-5}$ | $152 \pm 3$ | $80.5 \pm 4.0$ | $30.3 \pm 5.4$ |
| 10.0 | $(4.00 \pm 1.15) \times 10^{-5}$ | $288 \pm 3$ | $57.9 \pm 1.8$ | $20.2 \pm 3.5$ |

Obviously, the initial pH during UV irradiation of PFOA affected both decomposition kinetics and TPs formation. Most TPs were found for the UV treatment at pH 5.6, and the fewest for the treatment at pH 4.0, which only yielded PFHpA in a significant amount. By comparison, we determined and quantified PFHpA, PFHxA, PFPeA, and PFBA at pH 5.6. The choice of PFOA as a model compound can be justified because shorter-chain PFAAs are released as TPs during the decomposition of both PFOA and 6:2 FTAB (vide infra).

The release and amount of fluoride (F$^-$) as a major mineralisation product of PFOA seemed to be affected by pH also. The defluorination was almost comparable at acidic and neutral pH, averaging 30% after 360 min of treatment. However, at pH 5.6, F$^-$ release was first detectable after 30 min, while for pH 4.0 and pH 7.0, the F$^-$ release was detectable first after 50 min (Figure 1). At alkaline pH, the defluorination started even later, i.e., after 60 min from the beginning of the UV treatment, and it averaged 20% after 360 min (Table 1). However, our results indicate a two-stage decomposition reaction of PFOA, except at pH 4.0. At this pH value, produced protons are not quenched by OH$^-$, thus the reaction between protons and $e_{aq}^-$ reduces the availability of electrons that may be responsible for PFOA decomposition [39].

Our results clearly indicate a remarkable effect of the initial pH on the decomposition kinetics of PFOA and on the formation of the main TPs and major mineralisation product. Similar observations concerning the influence of pH were recently reported by Wang and Zhang, and others [37,38,41]. In particular, Wang and Zhang [37] investigated the influence of pH on PFOA decomposition, which was inhibited at alkaline pH. They suggested that the initial decomposition of PFOA is mediated by hydrated electrons ($e_{aq}^-$), generated through photolysis of OH$^-$ and dominating, at alkaline pH, over the direct PFOA photolysis. Since OH$^-$ ions produce $e_{aq}^-$, one could expect unaffected photolysis of PFOA. This assumption was also confirmed by Wang and Zhang [37], but only in the absence of oxygen. The influence of oxygen during PFOA photolysis was also recently investigated by Giri et al. [41]. Giri et al. and Wang and Zhang both observed negative effects of dissolved oxygen (DO) during PFOA photolysis, especially at alkaline pH. Wang and Zhang [37] postulated that, in the presence of oxygen at alkaline pH, $e_{aq}^-$ is scavenged by O$_2$ to produce O$_2^{\bullet-}$.

Recently, we demonstrated that our system configuration produces more O$_2^{\bullet-}$ than $^{\bullet}$OH radicals [51]. Therefore, the initial decomposition of PFOA should be significantly inhibited at alkaline pH in the presence of O$_2$. In our experimental set-up, we used deionised water without purging it with nitrogen gas, which caused DO to be present at the beginning of the UV treatment. Therefore, we can assume that, at alkaline pH, the presence of DO inhibits PFOA decomposition as postulated by Wang and Zhang [37] and Giri et al. [41].

Overall, our results are consistent with those of Wang and Zhang [37] and Giri et al. [41]. However, the two-stage decomposition reaction observed for PFOA in our study can also be possibly due to differences in the system configurations, leading to different kinetics involving H$_2$O$_2$ production as well as other factors (vide infra).

Our UV system configuration has tremendous impact on the decomposition kinetics and generation of radical species, as recently demonstrated [51]. Therefore, we are aware that the intensity of our UV lamp might also have an important effect on PFOA decomposition. Our UV light intensity was considerably weaker as compared to those

reported by Wang and Zhang and Giri et al. [37,41], resulting in lower photon flux and slower reaction kinetics. Therefore, determination of lower degradation rate constants is justified. Apart from these differences, Wang and Zhang [37] observed $H_2O_2$ formation from recombination of hydrogen peroxide radicals ($HO_2{}^\bullet$) in their system within 60 min, reaching to an optimum level of 300 µM between 20 min and 30 min. We also measured $H_2O_2$ formation in our system but at significantly lower concentration, close to the limit of detection.

As demonstrated by Wang and Zhang [37], it is very reasonable that high yields of $e_{aq}^-$ during oxygen free photolysis decompose PFOA. The presence of high quantities of oxygen will scavenge $e_{aq}^-$ and form $O_2{}^{\bullet-}$, especially at alkaline pH. In our system configuration, we measured DO consumption of 2 mg L$^{-1}$ within 180 min of UV-treated PFOA. Thus, we assume that rapid degradation of PFOA in the initial stage of our UV treatment was possible due to hampered $HO_2{}^\bullet$ formation.

*3.2. Scavengers' Experiments for PFOA Decomposition*

In our system configuration, we assumed that different reactive species, including $^\bullet OH$, $HO_2{}^\bullet$, and $O_2{}^{\bullet-}$, are synergistically participating in PFOA decomposition. To further investigate the mechanism potentially driven by different reactive species, scavenger experiments were carried out. Various scavengers including methanol (MeOH), ethanol (EtOH), 2-propanol, and t-butanol (for $^\bullet OH$), as well as L-Threoascorbic acid (for reactive oxygen species, ROS, including $O_2{}^{\bullet-}$, carbon dioxide radical, $CO_2{}^{\bullet-}$, and $^\bullet OH$ radical) were used [52–54].

We observed almost 90% PFOA decomposition without the addition of scavengers, as mentioned above. By adding different alcohols as $^\bullet OH$ scavengers, we always found inhibition of PFOA decomposition, lower efficiency, and lower defluorination, as compared with the reference treatment (Figure 2). Methanol, ethanol, and 2-propanol allowed for about 81–85% PFOA decomposition, while t-butanol gave only 75% decomposition (Table 2).

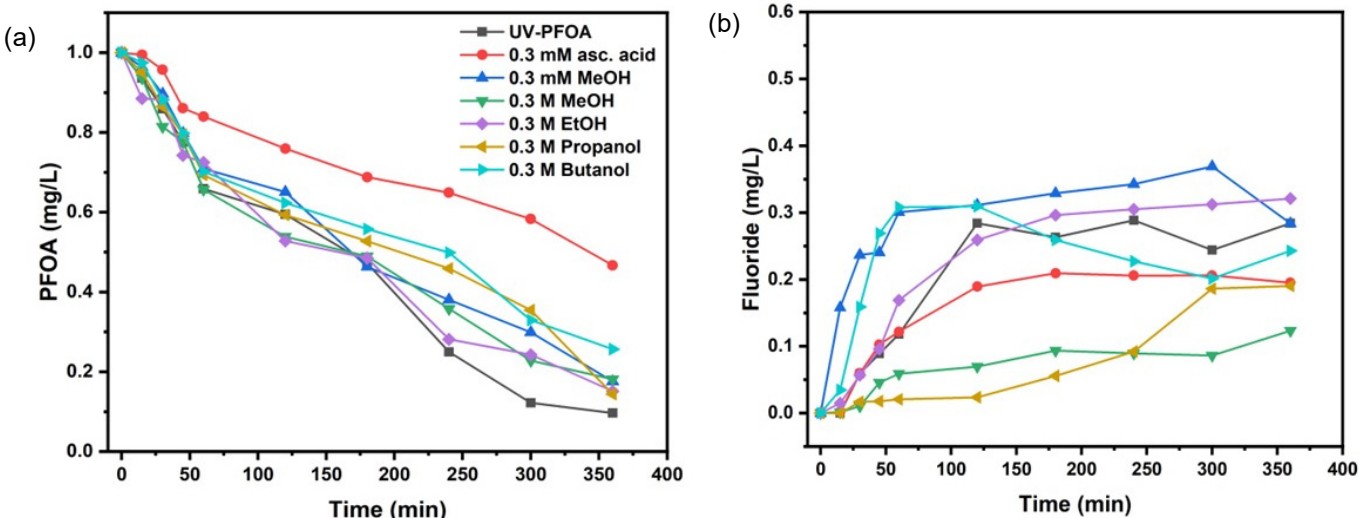

**Figure 2.** Comparative removal of PFOA in the presence of scavengers (**a**), fluoride release with scavengers' addition (**b**).

The alcoholic scavengers seemed to also affect the defluorination (Figure 2b). While the release of F$^-$ from PFOA without scavengers started only after 20 min, we found immediate release of F$^-$ in the presence of 0.3 mM MeOH. Overall, the defluorination with 0.3 mM MeOH was the highest (around 36% after 180 min treatment time, Table 2). Individual results of PFOA decomposition with all scavengers are shown elsewhere (Figure S3).

**Table 2.** Rate constant, half-life, %age decomposition, and defluorination by different scavengers for 1 mg L$^{-1}$ PFOA.

| Scavenger & Initial pH | k (s$^{-1}$) | t$_{1/2}$ (min) | Decomposition (%) | Fluoride Release (%) |
|---|---|---|---|---|
| 0.3 mM Methanol—pH-5.1 | $(8.06 \pm 1.5) \times 10^{-5}$ | $143 \pm 3$ | $82.6 \pm 2.9$ | $36.9 \pm 1.3$ |
| 0.3 M Methanol—pH-5.1 | $(7.91 \pm 0.2) \times 10^{-5}$ | $146 \pm 4$ | $81.9 \pm 2.7$ | $12.3 \pm 4.3$ |
| 0.3 M Ethanol—pH-5.7 | $(8.74 \pm 1.2) \times 10^{-5}$ | $132 \pm 5$ | $84.9 \pm 1.9$ | $32.1 \pm 3.2$ |
| 0.3 M Propanol—pH-5.8 | $(9.00 \pm 1.9) \times 10^{-5}$ | $128 \pm 5$ | $85.7 \pm 1.5$ | $19.0 \pm 2.8$ |
| 0.3 M Butanol—pH-5.7 | $(6.30 \pm 1.5) \times 10^{-5}$ | $183 \pm 3$ | $74.3 \pm 4.2$ | $30.8 \pm 4.6$ |
| 0.3 mM Ascorbic acid *—pH-4.3 | $(3.52 \pm 1.3) \times 10^{-5}$ | $327 \pm 6$ | $53.3 \pm 5.2$ | $20.9 \pm 4.2$ |

* at t = 0 and t = 1 h.

The addition of alcohols did not significantly affect the initial PFOA decomposition (Table 2). However, our results reveal that $^{\bullet}$OH radicals were being scavenged by alcohols as expected, but without inhibiting defluorination. This might be justified by the unrestricted presence of that which should be mainly responsible for PFOA decomposition [37,39,41].

Recently, Chen et al. [39] studied PFOA decomposition using a UV/H$_2$O/alcohol system. They observed that alcohols quench $^{\bullet}$OH radicals and, thus, might produce alcohol radicals. Both species are inefficient at decomposing PFOA. They stated that during the quenching of $^{\bullet}$OH radicals, more hydrated electrons are produced in the presence of alcohols. Therefore, they suggested that alcohols may act as catalysts for PFOA decomposition. However, very high alcohol concentrations (65 mM) were required to achieve significant PFOA decomposition. Beyond this amount, continuous addition of alcohols had no further effect on generation. In addition, it has been suggested that using alcohols caused an increase in the surface tension of 10 mg L$^{-1}$ PFOA, with better dispersion of PFOA on the surface and, thus, enhanced decomposition.

However, we used a ten-fold lower concentration of PFOA (1 mgL$^{-1}$) and did not observe better decomposition. Also, as stated in Section 3.1, due to the different system configuration and weaker UV intensity, our findings might not be consistent with Chen et al. [39]. Moreover, it has also been stated that alcohols protect electrons from quenching by oxygen and protons. Considering the O$_2^{\bullet-}$ radical as the driving factor for PFOA decomposition, we agree with Giri et al. [41] and Wang and Zhang [37], who observed the formation of O$_2^{\bullet-}$ radicals upon the quenching of electrons with DO. To further prove our assumption, we also applied ascorbic acid as an O$_2^{\bullet-}$ scavenger. Interestingly, we found the highest inhibition of PFOA decomposition (which reduced to 53% at 360 min treatment time, Figure 2a) with ascorbic acid, which also corresponded well with the inhibition of defluorination (Figure 2b). Obviously, ascorbic acid showed the strongest inhibition effect compared with all other scavengers applied.

Recently, Bai et al. [22] studied the effect of O$_2^{\bullet-}$ radicals on PFAS decomposition using a series of PFAAs. They demonstrated the involvement of O$_2^{\bullet-}$ radicals in PFAA decomposition both theoretically, using density functional theory (DFT), and experimentally. They measured the O$_2^{\bullet-}$ decay rates in the presence of PFAA, and considered the effect of solvation on O$_2^{\bullet-}$ reactivity. The possible mechanism was examined by DFT calculations, as well as the thermodynamic viability of the reaction pathway between an O$_2^{\bullet-}$ radical and C$_2$F$_5$CO$_2^-$. They concluded that the $\alpha$-C atom ($\Delta$G$_R^\circ$ = $-4.09$ kcal mol$^{-1}$) is attacked by O$_2^{\bullet-}$, causing the C–F bond to break. Despite these findings, Metz et al. [27] critically opposed the idea on the basis of recent results they obtained [28], and argued that they produced O$_2^{\bullet-}$ radicals by three different systems to verify the involvement of superoxides. In none of the systems were they capable of finding a correlation between O$_2^{\bullet-}$ formation and PFAS decomposition.

Interestingly, our results might support the hypothesis of O$_2^{\bullet-}$ participating in PFOA decomposition. Based on our results, we might agree with the finding of Bai et al. [22], although, as mentioned above, our findings might be caused by our system configuration, which favours the generation of O$_2^{\bullet-}$ over $^{\bullet}$OH. Nevertheless, we also support the claim of Metz et al., as we could not completely confirm the occurrence of O$_2^{\bullet-}$ only based on the

application of ascorbic acid as a scavenger. As mentioned above, ascorbic acid scavenges ROS, a rather broad variety of reactive oxygen species that includes, among others, $^{\bullet}OH$ and $O_2^{\bullet-}$.

Therefore, we have to address this important issue in future investigations, more precisely considering the given treatment conditions and parameters, to determine the exact role of different reactive species involved in the decomposition of PFOA. Independently, our aim was to investigate whether our system can reliably decompose PFOA with reproducible results, comparable to those reported in the recent literature and, ultimately, use the PFOA results to achieve decomposition of 6:2 FTAB.

### 3.3. Decomposition of 6:2 FTAB at Different pH Values

UV treatment of 6:2 FTAB was carried out for 360 min at four different initial pH values. The original pH of aqueous 6:2 FTAB solution was 6.5. During the UV treatment of 1 mg L$^{-1}$ 6:2 FTAB, complete decomposition was achieved within 360 min (Figure 3b). The decomposition followed more or less pseudo-first-order kinetics, with average lifetime of 45.7 ± 1.7 min at pH 6.5 (Table 3). The time evolution of 6:2 FTAB was actually characterised by a first exponential decay branch, resulting in a pseudo-plateau at 60–90 min that was followed by a second exponential decay, the outcome of which was complete 6:2 FTAB degradation at around 360 min (see Figure 3).

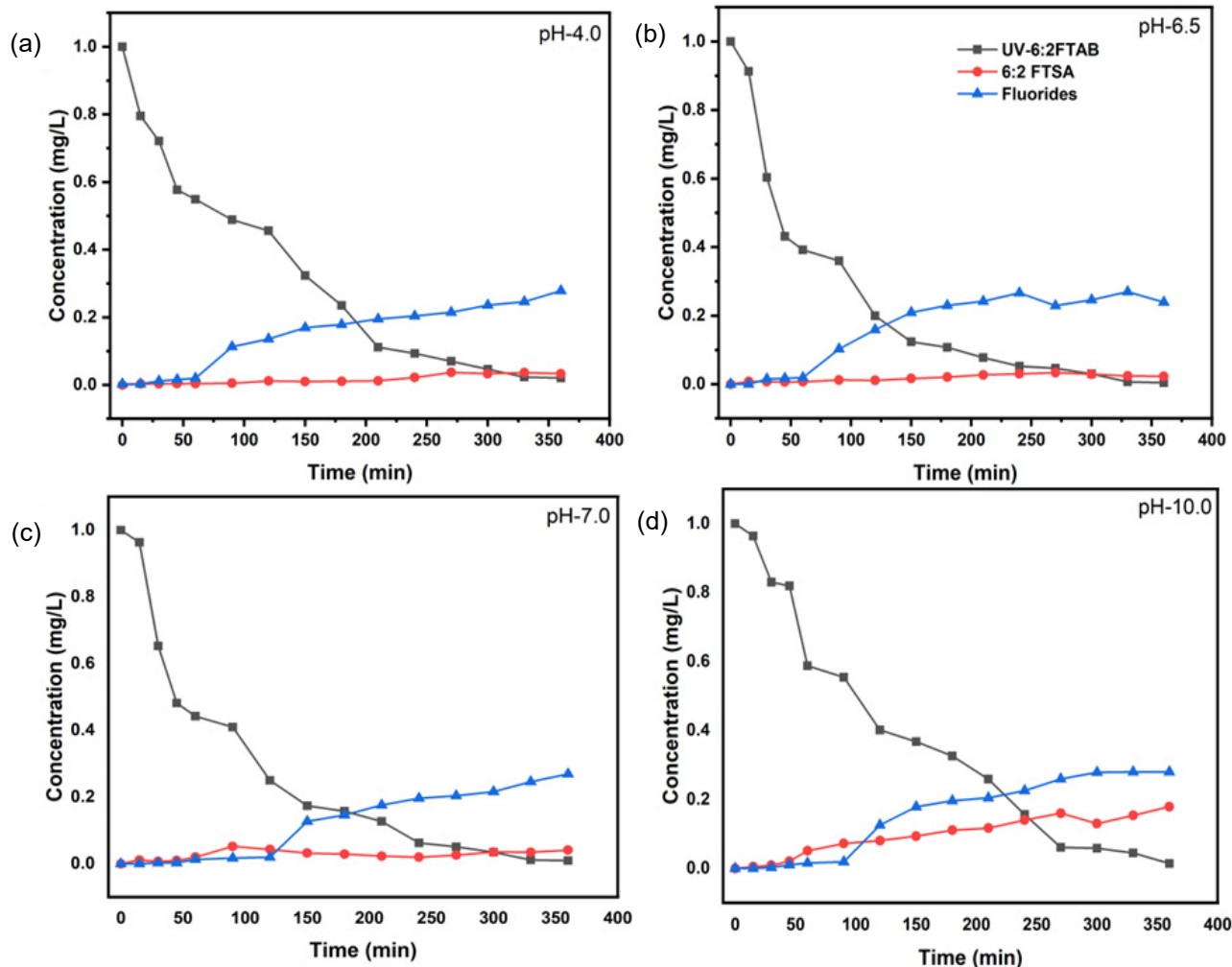

**Figure 3.** Removal of 6:2FTAB, formation of major transformation product, and fluoride release by UV irradiation at pH 4.0 (**a**), at pH 6.5 (**b**), at pH 7.0 (**c**), and at pH 10.0 (**d**).

**Table 3.** Rate constant, half-life, %age decomposition, and fluoride release at different pH for 1 mg L$^{-1}$ 6:2 FTAB.

| pH Value | k (s$^{-1}$) | t$_{1/2}$ (min) | 6:2 FTAB Decomposition (%) | Fluoride Release (%) |
|---|---|---|---|---|
| 4.0 | $(1.80 \pm 0.06) \times 10^{-4}$ | 64.1 ± 7.2 | 98.0 ± 3.1 | 27.9 ± 5.0 |
| 6.5 | $(2.52 \pm 0.16) \times 10^{-4}$ | 45.7 ± 1.7 | 99.6 ± 0.5 | 27.0 ± 4.7 |
| 7.0 | $(2.16 \pm 0.29) \times 10^{-4}$ | 53.3 ± 3.9 | 99.1 ± 0.2 | 26.9 ± 3.2 |
| 10.0 | $(1.97 \pm 0.14) \times 10^{-4}$ | 58.4 ± 6.2 | 98.6 ± 1.4 | 28.0 ± 4.2 |

The UV treatment of 6:2 FTAB was affected by pH as well. The lifetime followed the order pH 4 > pH 10 > pH 7, and the optimal pH for the UV degradation of 6:2 FTAB was 6.5 (Table 3). Despite the different half-lives determined, we always observed complete decomposition of 6:2 FTAB within 360 min of treatment time. In all cases, a two-branch bi-exponential time evolution curve could be observed (Figure 3).

Surprisingly, we only detected 6:2 fluorotelomer sulfonic acid (6:2 FTSA) as the major TP, with the highest concentration at alkaline pH. There is some evidence from the time evolution curves that the decomposition kinetics of 6:2 FTSA were slower at alkaline pH, where a slow accumulation of 6:2 FTSA occurred. Fluoride release started only after 60 min of treatment for all four different pH values (Figure 3b), and it amounted to 26–28% (Table 3). Some shorter-chain PFAAs were also identified, but only at the trace level, including PFHpA, PFHxA, and PFPeA. The rapid decomposition of 6:2 FTAB and low detection of TPs led us to assume that mineralisation to carbon dioxide ($CO_2$) might have taken place during the UV treatment. Unfortunately, our system configuration did not allow for the measurement of $CO_2$.

Many other studies reporting the decomposition of some telomer-related alcohols observed different major TPs, such as 6:2 fluorotelomer sulfonic acid (6:2 FTSA), 6:2 fluorotelomer alkyl acid, 6:2 FTAA, 6:2 FTSAm, and others (6:2 FTCA, 6:2 FTUCA, etc.) [55–60]. It is surprising that we did not detect more TPs during 6:2 FTAB decomposition in our treatment, which might be due to kinetic effects.

As mentioned above, the decomposition of 6:2 FTAB in our treatment system followed a two-stage kinetics. In the initial stage of the UV treatment, direct photolysis of 6:2 FTAB might be reasonable. However, the participation of reactive species such as hydroxyl radicals and hydrated electrons cannot be discussed here, without further information on their presence in the UV system.

*3.4. Scavengers Experiments for 6:2 FTAB Decomposition*

To study the involvement of reactive species in the decomposition of 6:2 FTAB by photolysis, the same scavengers as for PFOA were used to better understand the role of different reactive species (Figure 4 and Table 4). The application of different types of scavengers inhibited the UV decomposition of 6:2 FTAB. All scavengers had immediate inhibiting effects, except for MeOH. For either 0.3 M or 0.3 mM MeOH, the inhibition effect only occurred 30 min after the start of the UV treatment.

As mentioned above, alcohols are $^\bullet$OH scavengers. To some extent, ascorbic acid can also inhibit the reaction by scavenging both $O_2^{\bullet-}$ and $^\bullet$OH or related ROS (Figure 4). The exception of 0.3 mM methanol as a scavenger could be explained based on its lower concentration as compared to other alcohols, typically used at 0.3 M concentration.

Interestingly, $^\bullet$OH was shown to play little to no role in the degradation of PFOA, but the hydroxyl radical was directly involved in the 6:2 FTAB degradation mechanism. In this framework, 0.3 mM MeOH is likely to be a poor $^\bullet$OH scavenger and, coherently, it inhibited 6:2 FTAB decomposition to only a limited extent (Figure 4a). We also found that alcohols with a longer carbon chain caused a higher and more significant inhibition of 6:2 FTAB degradation.

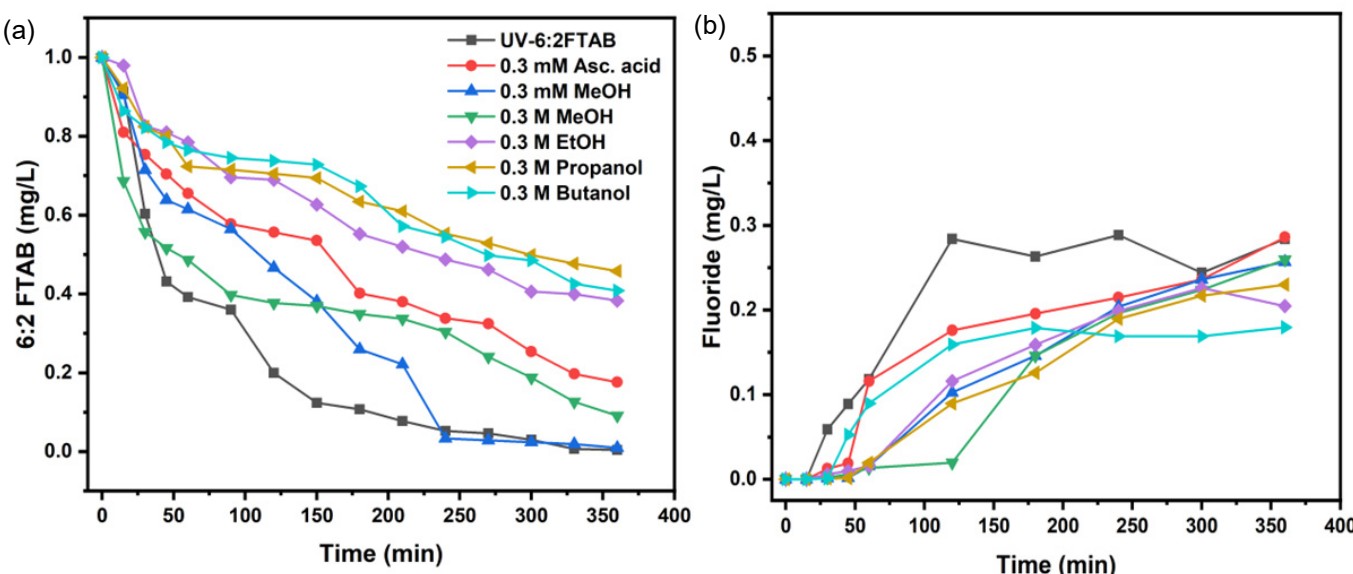

**Figure 4.** Decomposition of 6:2 FTAB in the presence of scavengers (**a**), fluoride release with scavengers' addition (**b**).

**Table 4.** Rate constant, half-life, %age decomposition, and fluoride release with different scavengers for 1 mg L$^{-1}$ 6:2 FTAB.

| Scavengers | k(s$^{-1}$) | t$_{1/2}$ (min) | 6:2 FTAB Decomposition (%) | Fluoride Release (%) |
|---|---|---|---|---|
| 0.3 mM Methanol | $(2.15 \pm 0.12) \times 10^{-4}$ | $54 \pm 4$ | $99.0 \pm 0.7$ | $25.7 \pm 4.4$ |
| 0.3 M Methanol | $(1.10 \pm 0.24) \times 10^{-4}$ | $104 \pm 8$ | $90.9 \pm 4.3$ | $26.0 \pm 2.6$ |
| 0.3 M Ethanol | $(4.44 \pm 0.41) \times 10^{-5}$ | $260 \pm 7$ | $61.7 \pm 2.9$ | $22.6 \pm 3.7$ |
| 0.3 M Propanol | $(3.61 \pm 0.32) \times 10^{-5}$ | $319 \pm 7$ | $54.2 \pm 2.6$ | $23.0 \pm 2.0$ |
| 0.3 M Butanol | $(4.15 \pm 0.97) \times 10^{-5}$ | $279 \pm 10$ | $59.2 \pm 3.5$ | $18.0 \pm 4.3$ |
| 0.3 mM Ascorbic acid * | $(8.04 \pm 0.55) \times 10^{-5}$ | $144 \pm 8$ | $82.4 \pm 4.2$ | $25.6 \pm 2.9$ |

\* at t = 0 and t = 1 h.

Interestingly, 0.3 mM ascorbic acid inhibited 6:2 FTAB decomposition to a higher extent than either 0.3 mM or 0.3 M MeOH, but less than EtOH, 2-propanol, and t-butanol. By the addition of 0.3 mM ascorbic acid, a more inhibited effect was observed than MeOH, which indicates that both hydroxyl and superoxide radicals are being scavenged by ascorbic acid, so a greater effect of inhibition was observed in the case of ascorbic acid. All scavengers decreased fluoride release that ranged between 22 and 28%, except for t-butanol, which reduced F$^{-}$ release down to 17%. The SO$_4^{2-}$ release was observed most often at higher pH, while at lower and neutral pH values, it was not observed due to low resolution.

Although the addition of ascorbic acid also inhibited the decomposition of 6:2 FTAB, the scavenging effect was not as strong as with EtOH, 2-propanol, or t-butanol. These findings lead us to assume that the decomposition mechanism of 6:2 FTAB is an overlapping process of direct and indirect photolysis, driven by either/both hydrated electrons and/or radicals.

The complex role of different reactive species participating in the decomposition of 6:2 FTAB was also recently described by Trouborst [58]. He studied 6:2 FTAB photolysis in a photoFate system, involving sunlight for decomposition, obtaining different TPs including mainly 6:2 FTSAm, 6:2 FTSA, and some short-chain PFAAs in very low concentration. Trouborst explained the formation of 6:2 FTSAm mainly by direct photolysis, but in our UV treatment system, we did not observe the formation of 6:2 FTSAm. We assume that if 6:2 FTSAm were released, it might have volatilised due to the open system configuration. In the study by Trouborst [58], release of 6:2 FTSAm might be possible due to sunlight absorption

by 6:2 FTAB. However, there were many degradation products, ultimately degraded to short-chain PFAAs. The treatment lasted from many hours to days and the experimental set-up was complicated. Moreover, the formation of 6:2 FTSA can be due to subsequent degradation of 6:2 FTSAm, but also to direct photolysis from 6:2 FTAB, as we assume from our results. It has been reported [61] that the biodegradation of 6:2 FTSA also produces 6:2 fluorotelomer carboxylic acid (6:2 FTCA), 6:2 fluorotelomer unsaturated carboxylic acid (6:2 FTUCA), and some short-chain PFAAs [59,61]. Regarding the biodegradation of 6:2 FTSA, slower and incomplete degradation was observed. We also observed the formation of some short-chain PFAAs, including PFHpA, PFHxA, and PFPeA, but only at trace levels. The individual effect of scavengers over 6:2 FTAB removal, along with 6:2 FTSA as the major TP and fluoride release, is shown elsewhere (Figure S4).

### 3.5. Proposed Mechanism of 6:2 FTAB Decomposition

Based on our results, obtained from UV treatments and scavenger experiments, we propose a mechanism for the UV decomposition of 6:2 FTAB, assuming that the process is mediated by both direct and indirect photolysis. In other words, we assume that UV radiation and $^{\bullet}OH$ play the main roles (Figure 5), which is supported by experimental results including $SO_4^{2-}$ release.

**Figure 5.** Proposed UV photolysis pathways of 6:2 FTAB. Blue arrows represent the major observed decomposition pathway, hypothesised according to the transformation products detected in this study. Red arrows indicate other reasonable decomposition routes [56,58,59,62].

The formation of 6:2 FTSA can be explained by S-N bond breaking by UV radiation (step 1), followed by $^{\bullet}OH$ attack on the sulphur atom. We observed the formation of 6:2

FTSA in every sample, but did not observe either 6:2 FTAA or 6:2 FTSAm. The formation of 6:2 FTSA via these metabolites, as well as directly from the mother compound are both feasible [58]. The TP 6:2 FTSA would further undergo $^\bullet$OH-mediated reactions (step 4 and onwards) to form short-chain PFAAs (PFHpA, PFHxA, PFPeA, etc). Yang et al. [56] also studied the decomposition of 6:2 FTSA using $UV/H_2O_2$ and proposed a mechanism involving $^\bullet$OH attack as a prerequisite to mineralisation. Furthermore, by removing $CF_2$ groups, shorter-chain PFAAs are released [56,58,62]. Beatriz et al. [63] studied the electrochemical decomposition of 6:2 FTAB, also observing 6:2 FTSA release and the formation of short-chain PFAAs, similar to our results. They also proposed the direct formation of 6:2 FTSA from the mother compound and via other metabolites.

## 4. Conclusions

We investigated the UV photolysis of PFOA and of 6:2 FTAB at different pH values (4.0, 7.0, 10.0, and the original pH, respectively, 5.6 and 6.5). All UV treatments lasted for more than 300 min. PFOA decomposition yielded shorter-chain PFAAs like PFHpA, PFHxA, and PFPeA as major TPs. The process was most effective at pH 5.6; moreover, acidic pH had only minor effects on PFOA decomposition, while basic pH provided considerable inhibition. The decomposition of 6:2 FTAB was affected by pH as well, and it yielded some shorter-chain PFAAs and 6:2 FTSA, that were identified by LC-MS analysis.

The results of scavenger experiments indicated that alcohols did not significantly affect the decomposition of PFOA, whereas ascorbic acid carried out considerable inhibition. Considering the roles of alcohols as selective $^\bullet$OH scavengers and of ascorbic acid as a general ROS scavenger (including $^\bullet$OH), these results suggest that $^\bullet$OH was not involved in the decomposition of PFOA, which likely proceeded with the contribution of $e_{aq}^-$ and/or $O_2{}^{\bullet-}$.

As far as 6:2 FTAB decomposition is concerned, the major decomposition product 6:2 FTSA was likely formed via the involvement of both direct and indirect photolysis. Differently from PFOA, we found clear indication that both $^\bullet$OH (especially) and $O_2{}^{\bullet-}$ are involved in the decomposition of 6:2 FTAB.

In particular, the role of $O_2{}^{\bullet-}$ could not be elucidated clearly in this study and needs to be addressed in future work. Moreover, clarification is needed regarding the role of $e_{aq}^-$ in the degradation of 6:2 FTAB in future work. The efficiency of the treatment might be further improved by introducing stronger UV sources (for example, vacuum UV), suitable sensitisers, and optimised pH and temperature throughout the treatment.

**Supplementary Materials:** The following supporting information can be downloaded at: https://www.mdpi.com/article/10.3390/chemengineering8020032/s1.

**Author Contributions:** Conceptualisation, N.A., R.R. and M.M.; methodology, N.A., R.R. and M.M.; validation, N.A. and R.R.; investigation, N.A., R.R. and M.M.; data curation, N.A., R.R. and M.M.; writing—original draft preparation, N.A. and R.R.; writing—review and editing, N.A., R.R., I.M.B., D.V. and M.C.B.; visualisation, N.A.; supervision, M.M. and R.R. All authors have read and agreed to the published version of the manuscript.

**Funding:** MCB and DV acknowledge financial support by Project CH4.0 under the MIUR program "Dipartimenti di Eccellenza 2023–2027" (CUP: D13C2200352001).

**Data Availability Statement:** The data that support the findings of this study are available on request from the corresponding author.

**Acknowledgments:** The first author acknowledges Martienssen for the welcoming at BTU Cottbus-Senftenberg, for invaluable guidance, and to the laboratory team for their unwavering support during the research.

**Conflicts of Interest:** The authors declare no conflicts of interest.

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
