# Peer review of "Investigation on UV Degradation and Mechanism of 6:2 Fluorotelomer Sulfonamide Alkyl Betaine, Based on Model Compound Perfluorooctanoic Acid"

_2305-7084, doi:10.3390/chemengineering8020032_

Round 1

Reviewer 1 Report

Comments and Suggestions for Authors

Investigation on UV degradation and mechanism of 6:2 fluorotelomer sulfonamide alkyl betaine, based on model compound perfluorooctanoic acid (Chem Engineering-2824275)

Comments: The degradation of perfluorooctanoic acid (PFOA) is a great challenge with low efficiency and high operating cost. In order to solve this issue, many studies have tried to use electrochemical oxidation and generation degradation methods, with good results. But in the actual treatment, a high chemical dosage and a high cost shortage limited its further application. Recent studies have proposed some methods, namely, the hydrothermal method + alkali treatment for UV photodegradation. In this study, the authors focused on the UV photodegradation of 6:2 fluorocarbon sulfonamide alkyl betaine based on the degradation process of the model compound and some of the related mechanisms. During PFAS degradation process, the mechanism of direct photolysis and electron reaction with hydration was proposed, excluding the involvement of hydroxyl radicals. At present, the role of superoxide radicals is uncertain. This topic is meaningful for the degradation of PFAS, but in-depth mechanistic studies and related discussions are not detailed enough.

Several issues should be clarified in this manuscript.

(1) Abstract: The background of the relevant 6:2 sulfonane alkyl UV degradation treatment should be mentioned.

(2) Abstract: The main UV conversion product was identified as 6:2 fluoroalkyl sulfonic acid (6:2 FTSA) with short chain perfluoroalkyl acid (PFAA), and other exercise products should also be presented.

(3) Introduction: Please describe the reasons for the choice of PFOA and 6:2 FTAB as research models and explain why both are important for environmental and health issues.

(4) Introduction: Knowledge gaps of 6:2 FTAB degradation using UV should be well organized.

(5) Materials and methods: The purpose of the free radical scavenging experiment is not explicitly given.

(6) Results and Discussion: The incorporation of ascorbic acid impeded the decomposition of 6:2 FTAB, with 0.3 mM of ascorbic acid exhibiting greater restraint than both 0.3 mM and 0.3 M methanol, albeit less effective than EtOH, 2-propanol and t-butanol. The role of ascorbic acid is not included here.

(7) Results and Discussion: When comparing with the results of other studies, the authors should further discuss the advantages and disadvantages.

(8) ConclusionAll UV treatments lasted more than 300 min, the experiment lasted too long and was expected to be improved. Please discuss its further solving solutions for improving its efficiency.

Author Response

On behalf of the authors, I would like to thank you very much for reviewing our manuscript so very precise and detailed. With most of your suggestions, recommendations and comments we completely agree and improved our manuscript accordingly. Please find our complete response below.

Reviewer 2 Report

Comments and Suggestions for Authors

The article titled “Investigation on UV degradation and mechanism of 6:2 fluorotelomer sulfonamide alkyl betaine, based on model compound perfluorooctanoic acid” focuses on the degradation of perfluorooctanoic acid (PFOA) and 6:2 fluorotelomer sulfonamide alkyl betaine (6:2 FTAB, Capstone B), using UV photolysis under various pH conditions. Although the study objective is mentioned, still some points should be clarified:

1.     For the decomposition of PFOA at different pH values, why pH levels (4.0, 5.6, 7.0, 10.0) were chosen. For example, what is the points of zero charge (PZC)?

2.     The proposed mechanisms of PFOA and 6:2 FTAB decompositions need clarification (Figure 5 should be revised).

3.     What about the generated byproducts?

4.     What are the optimum operational factors (e.g., pH, time, etc.)

5.     What are the kinetic constants of the decomposition rates

6.     The abstract should show the research gap to be discovered?

7.     In the last paragraph of the Introduction, mention the study objectives.

8.     The data should be supported by statistical analysis such Tukey post-hoc test

9.     What is the correlation between the study outputs and sustainability?

10.  In the conclusion section, mention the significant findings and future recommendations

11.  The data in figures and tables should be represented with statistical ranges (min, max, stdv., average)

Comments on the Quality of English Language

Moderate editing of English language required

Author Response

Complete response letter to the comments of reviewer 2

Dear reviewers,

Hereby we response to your comments on the manuscript:

“Investigation on UV degradation and mechanism of 6:2 fluorotelomer sulfonamide alkyl betaine, based on model compound perfluorooctanoic acid”

Naveed Ahmed, Marion Martienssen, Isaac Mbir Bryant, Davide Vione, Maria Concetta Bruzzoniti, Ramona Riedel

On behalf of the authors, I would like to thank you very much for reviewing our manuscript so very precise and detailed. With most of your suggestions, recommendations and comments we completely agree and improved our manuscript accordingly. Please find our complete response below.

Yours sincerely,

Ramona Riedel                                                                                      Cottbus, 27th January 2024

Dr.-Ing. Ramona Riedel  (Ph.D.)
Brandenburg University of Technology
Biotechnology of Water Treatment
Siemens-Halske-Ring 8
03046 Cottbus,

Germany
phone: +49 355 694308

Answer as follows:

Comments by Reviewer #2

  1. For the decomposition of PFOA at different pH values, why pH levels (4.0, 5.6, 7.0, 10.0) were chosen. For example, what is the points of zero charge (PZC)?

Dear reviewer 2, an explanation for choosing these specific pH values was already highlighted in line 213 to 216.

  1. The proposed mechanisms of PFOA and 6:2 FTAB decompositions need clarification (Figure 5 should be revised).

We carefully revised Figure 5.

  1. What about the generated byproducts?

Dear reviewer, we understand your comment regarding both PFOA and 6:2 FTAB. We hope that is right, because your question was unprecise. The generated by-products are further mineralized to water, carbon dioxide, and fluoride in the case of PFOA while in the case of 6:2 FTAB, by-products are mineralized water, carbon dioxide, fluoride, and sulphate ions. However, to demonstrate final mineralisation longer UV treatment was required up to 24 to 48 h. We avoided such long treatments due to protect the lifetime of our UV lamp.

  1. What are the optimum operational factors (e.g., pH, time, etc.)

For PFOA, pH value of 5.6 with a treatment time of 360 minutes optimal UV decomposition will be achieved up to 90% decomposition as observed. For 6:2 FTAB, all pH values investigated result in complete removal within 360 min of UV treatment as demonstrated in our manuscript.

  1. What are the kinetic constants of the decomposition rates

Dear reviewer 2, all kinetic degradation data are already given in Table 1 to 4.

  1. The abstract should show the research gap to be discovered.

We added the missing information in the abstract accordingly to line 15 to 17.

  1. In the last paragraph of the Introduction, mention the study objectives.

Dear reviewer 2, the objective was already given in the last paragraph of the introduction.

  1. The data should be supported by statistical analysis such Tukey post-hoc test

We agree that there should be some statistical analysis. We apologize for not performing these statistical tests as we believe, for now, it would take much time and would be difficult to perform these analyses.

  1. What is the correlation between the study outputs and sustainability?

The UV treatment option and parameters considering the persistency and toxicity of PFOA and 6:2 FTAB are promising to be considered for their effective removal from aqueous media. PFAS can be removed effectively using the given outputs.

  1. In the conclusion section, mention the significant findings and future recommendations

Significant findings are already in the conclusion section relevant to PFOA and 6:2 FTAB paragraphs. Future recommendations are added between lines 515 and 520.

  1. The data in figures and tables should be represented with statistical ranges (min, max, stdv., average).

We agree with your kind suggestion. However, we believe adding the relevant required information in the figures would overload the graphs.

Reviewer 3 Report

Comments and Suggestions for Authors

Accepted with minor changes. 

This manuscript delivers detailed research on the UV photolysis degradation of perfluorooctanoic acid (PFOA) and 6:2 fluorotelomer sulfonamide alkyl betaine, which are toxic to human’s health and environment. The degradation efficiency was evaluated under various pH environments, ranging from 4 to 10. The best degradation pH value is 5.6, which a 90% decomposition could be achieved after 360 minutes. Additionally, they also proposed a mechanism toward the photolysis process and provide their discussion and understanding. Additionally, the manuscript is effectively composed and easily comprehensible. In conclusion, I recommend accepting this manuscript after an improvement toward the figure 5. In figure 5, the angle and direction of bonds and atoms are stretched and a bit distorted. So It would be better that the authors can modify the structures in a more reasonable and good-looking way.

Author Response

On behalf of the authors, I would like to thank you very much for reviewing our manuscript so very precise and detailed. With most of your suggestions, recommendations and comments we completely agree and improved our manuscript accordingly. Please find our complete response below.

We improve figure 5 and added a new version to the manuscript.

Round 2

Reviewer 1 Report

Comments and Suggestions for Authors

The manuscript was revised according to the reviewers comments, so that I recommended an acceptance.

Comments on the Quality of English Language

Can be readable. 

Reviewer 2 Report

Comments and Suggestions for Authors

The authors responses to my comments are satisfactory.

Comments on the Quality of English Language

Minor editing of English language required